



# A Study on the Fully Coupled Atmosphere-Land-Hydrology Process and Streamflow Simulations over the Source Region of the Yellow River

Yaling Chen[1], Jun Wen[1], Xianhong Meng[2], Qiang Zhang[1], Xiaoyue Li[1], Ge Zhang[1]

[1]Key Laboratory of Plateau Atmosphere and Environment, Sichuan Province, College of Atmospheric Sciences, Chengdu University of Information Technology, Chengdu 610225, China

[2]Key Laboratory of Land Surface Process and Climate Change in Cold and Arid Regions, Northwest Institute of Eco-Environment and Resources, Chinese Academy of Sciences, Lanzhou 730000, China

*Correspondence to*: Jun Wen (jwen@cuit.edu.cn)

**Abstract.** The Source Region of the Yellow River (SRYR) is known as the "Water Tower of the Yellow River", which is the most important water conservation area in the upper reaches of the Yellow River. The streamflow of the SRYR makes an important contribution to the water resources in the Yellow River basin. Based on the Weather Research and Forecasting Model Hydrological modeling system (WRF-Hydro) model, by using meteorological, hydrological observations and reanalysis data, the key variables of the coupled atmosphere-land-hydrological processes over the SRYR during the 2013 rainy season (May-August) are analyzed, and the simulation results of the fully coupled WRF-Hydro with those of the standalone WRF are compared, whose aim is to assess the impact of hydrological coupling on the regional atmospheric model settings. The results show that the WRF-Hydro model has ability to depict the characteristics of streamflow over the SRYR with a Nash Efficiency Coefficient (NSE) of 0.44 during the calibration period from June 1[st] , 2012 to September 30[th], 2012 and a NSE of 0.61 during the validation period from May 1[st], 2013 to August 31[st], 2013. Compared with the standalone WRF model, the fully coupled model tends to show better performance with respect to temperature, downward longwave radiation, downward shortwave radiation, latent heat, sensible heat and soil temperature and moisture. Although the wet bias of the coupled simulated precipitation slightly increases (2.51 mm vs. 2.50 mm) due to the consideration of lateral flow of soil water, the simulation results of the land-atmosphere water-heat exchange fluxes and soil heat fluxes are comparably improved. Compared with the observations, the mean Root Mean Square Error (RMSE) of latent and sensible heat is reduced to 32.27 W·m$^{-2}$ and 24.91 W·m$^{-2}$, and of surface soil temperature and moisture is reduced to 4.22 K and 0.06 m$^3$/m$^3$. Besides, the fully coupled model is able to capture the variation characteristics of streamflow with a NSE of 0.33, which indicates that the fully coupled WRF-Hydro model has great potential for characterizing coupled atmosphere-land-hydrological processes and streamflow simulation in the cold climatical and complex topographic regions.

## 1 Introduction

Water, energy and heat flux and the processes among the atmosphere, land surface and hydrology are closely linked through highly complex interactions (Arnault et al., 2016; Fersch et al., 2020). In the water cycle of the whole Earth Climate System, the land surface hydrological processes are the link between atmospheric water (e.g. precipitation, evapotranspiration, water



vapor transport), terrestrial surface water (e.g. rivers, lakes, glacial meltwater, snow meltwater, surface runoff, oceans),
groundwater (e.g. baseflow, subsurface runoff, soil water), and ecological water (vegetation water), which is able to provide
feedback to weather and climate by regulating land-atmosphere energy and water cycle processes (Zheng et al., 2020).
Therefore, understanding the hydrological cycle processes between the atmosphere and the land surface in the mesoscale
river basins is of great importance for local ecological protection and macro regulation of water resources (Milly et al., 2005).
The Source Region of the Yellow River (SRYR) is located in the hinterland of the Tibetan Plateau (TP). It belongs to the
continental semi-arid climate zone of the plateau, with complex climatic conditions and the temperature rise rate of
0.48 °C/(10a)$^{-1}$ and precipitation increase of 7.6 mm/(10a)$^{-1}$ (Meng et al., 2020). The region is sensitive and fragile to climate
change and ecological environment, with a large number of alpine lakes, wetlands, which is the "sensitive area" and "start-up
area" for East Asia and even global climate change (Wu et al., 2004). The SRYR is known as "Yellow River Water Tower"
and accounts for about 16.2% of the total area of the Yellow River Basin. The streamflow is dominated by precipitation and
glacial meltwater and is the main flow-producing area and water conservation area of the middle and upper reaches of the
Yellow River (Zheng et al., 2007). Historic records prove that the Yellow River civilization has been able to continue for
thousands of years, one of the important reasons is that there is a stable ecological environment and water supply of the
SRYR (Zhang et al., 2017). However, in recent years, with the increasing influence of global climate change and human
activities, as well as the uneven distribution of regional water resources, water conservation units such as glaciers,
permafrost and grasslands have experienced significant changes, extremely meteorological and hydrological events such as
rainstorm, blizzard, droughts and floods have occurred frequently, the spatiotemporal distribution of precipitation and
hydrological uncertainties of watersheds have also increased. The sustainable development of ecological environment and
social economy over the SRYR is confronted with great challenges (Milly et al., 2002; Yuan et al., 2018).
With the rapid development of high-resolution Earth System Models, the role of land surface variability on simulation results
is more and more emphasized (Clark et al., 2015; Tang et al., 2019). Currently, most researches on fully coupled
atmosphere-hydrology process use regional climate models (RCMs) or land surface models (LSMs) combined with
hydrological models to investigate the relationship between climate change and hydrological cycle processes (Kruk et al.,
2012). The main technical tools applied include satellite remote sensing, data assimilation, error correction, downscaling
(statistical downscaling and dynamic downscaling) analysis. The main models employed are the Weather Research and
Forecasting Model (WRF), Community Land Model (CLM), Community Noah Land Surface Model with Multi-
Parameterization Options (Noah-MP), and Soil and Water Assessment Tool (SWAT), with a focus on the impact of climate
change and human activities on hydrological cycle processes (Cuo et al., 2013; Sheng et al., 2017; Zheng et al., 2018).
However, most studies focus on the influence of climate change on the hydrological processes in watersheds, and adopt a
single-directional linkage of "atmospheric circulation change-regional precipitation change-land hydrological change",
which cannot accurately describe the feedback of land surface and hydrological processes on the regional climate and affect
the simulation accuracy of the hydrological processes in watersheds (Wen et al., 2011).



The Weather Research and Forecasting Model Hydrological modeling system (WRF-Hydro) is a high-resolution distributed
land-atmosphere coupled model developed by the National Center for Atmospheric Research (NCAR) to improve the
redistribution of surface, subsurface and river water and to facilitate the coupling of atmospheric and hydrological models
(Gochis et al., 2020). The WRF-Hydro model is able to be run either as a standalone land surface hydrological model or
coupled with an atmospheric model (such as WRF) to achieve a two-way feedback process between the atmosphere and land
surface. Compared to traditional land surface hydrological models, the WRF-Hydro model is designed to provide continuous
spatially gridded information on soil temperature and moisture, evapotranspiration, water and heat exchange fluxes, and
runoff (Gharamti et al., 2021; Gu et al., 2021). Presently, the WRF-Hydro model has been successfully applied to many
fully coupled atmosphere-hydrology studies (Zhang et al., 2019; Fersch et al., 2020; Eidhammer et al., 2021). Senatore et al.
(2015) found that the precipitation, surface runoff and surface fluxes simulated by the fully coupled WRF-Hydro model are
better than WRF model in the Crati River Basin. Li et al. (2021) concluded that the coupled WRF-Hydro model improved
the simulation results of soil moisture and precipitation, and has some potential in simulating and projecting of streamflow
over the Source Region of the Three River.
The above studies indicate that WRF-Hydro model has a wide range of applications and a strong capability in streamflow
simulation, so it has a great potential in climate and hydrology coupled simulation over the SRYR with complex underlying
surface conditions. In this research, the fully coupled WRF-Hydro model is used to simulate the rainy season (May-August)
of the Yellow River Source basin in 2013, and simulation results are compared to those of the standalone WRF model. The
focus is on the analysis of the differences caused by WRF and fully coupled WRF-Hydro for several variables directly
related to terrestrial water cycle processes, especially the effects of these variables on precipitation. Materials and
methodology are arranged in the section 2 and 3 respectively. The comparison between the simulation performance of the
standalone WRF and the coupled WRF-Hydro are followed. And then, the characteristics of fully coupled streamflow are
analyzed. Finally, the discussion and main conclusions are provided in section 5 and section 6.
**2 Study area and data**
**2.1 Study area**
The SRYR (32.12°-35.48°N, 95.50°-103.28°E) lies in the northeast of the TP, with the total catchment area and mean
elevation being 1.22×105 km2 and 4000.0 m respectively, as displayed in Fig. 1. This also constitutes the setup of the WRF
and fully coupled WRF-Hydro nested domain. The region is located in the edge area affected by the East Asian monsoon
and belongs to the plateau cold climate zone with an annual-mean temperature close to 0.0 ℃ and the annual-mean
precipitation between 300.0 and 500.0 mm (Ji et al., 2020). As the main flow-producing area and water-conserving area in
the middle and upper reaches of the Yellow River, the SRYR is known as the "Yellow River Water Tower", taking the
Tangnaihai hydrological station as the outlet of the basin (Zheng et al., 2007). It is mainly composed of high mountains,
plains and hills, and is distributed with vast lakes, including the largest plateau fresh water lakes in China, namely the





Zhaling Lake and the Eling Lake (Wen et al., 2011). The terrain is undulating with a relatively poor capacity for water
holding on the surface, and the main land use type is alpine meadow grassland and alpine wetland. Under the influence of
hydrological environment, the soil types are mainly composed of loam and sandy loam with rough texture, and the seasonal
frozen soil is widely distributed.

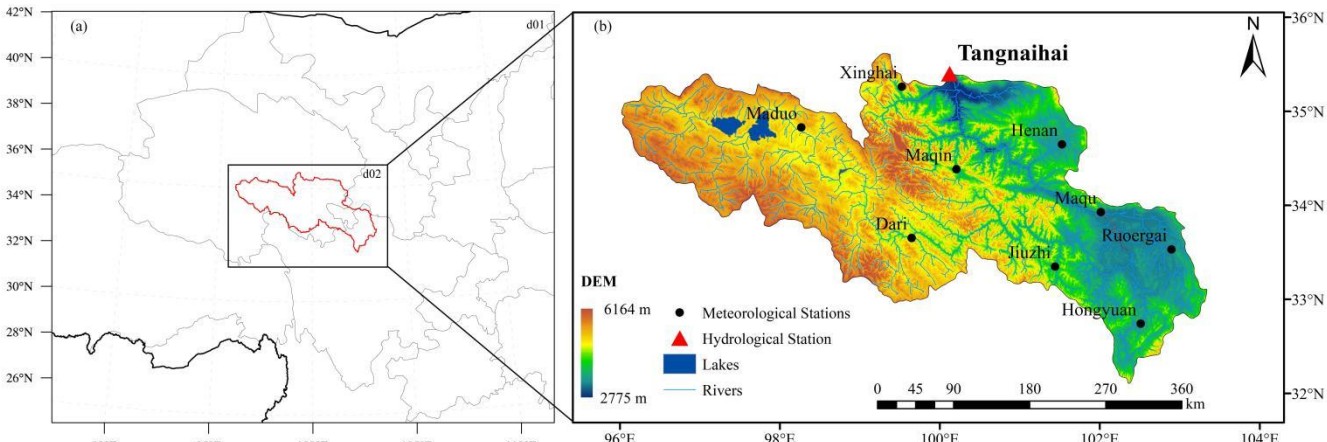

**Figure 1.** The WRF and WRF-Hydro nested domain (a), and the geographic locations of the meteorological and
hydrological stations of the studied catchments (b).
**2.2 Data**
The daily streamflow data of Tangnaihai hydrological station provided by Yellow River Water Conservancy Bureau during
2012-2013 are employed in this research. Besides, the turbulent heat fluxes and top-layer soil temperature and moisture data
of the Eling Lake station (Meng and Lyu, 2022) and Maqu station (An et al., 2019; 2020) from the National Cryosphere
Desert Data Center (http://www.ncdc.ac.cn) and the observations of 9 meteorological stations over the SRYR (Fig. 1b)
collected from the National Meteorological Information Center of China are also applied.
WRF-Hydro model requires a lot of input data, including meteorological driving data, underlying surface data and river
network data. The meteorological driving data are mainly composed of seven variables: downward longwave and shortwave
radiation, surface pressure, specific humidity, air temperature, near surface wind speed and precipitation rate. As the SRYR
is located in the hinterland of the TP, the meteorological observation stations are rare and unevenly distributed, which brings
great difficulties in driving model. The Global Land Data Assimilation System (GLDAS) data are jointly developed by
National Aeronautics and Space Administration (NASA) and the National Center of Environmental Prediction (NCEP) with
a temporal resolution of three hours and a spatial resolution of $0.25° \times 0.25°$, which integrate the ground and satellite
observation data and have great applicability over the SRYR (Li et al., 2014).
In addition, the quality of precipitation data is critical to streamflow simulation and is the most sensitive factor affecting the
variation of streamflow, so selecting a high-quality precipitation product as the precipitation driving field of WRF Hydro
model is of great significance. The China Meteorological Forcing Dataset (CMFD) is a high spatial-temporal resolution



(0.10°×0.10°) gridded meteorological driving dataset, which is developed for studies of land surface processes in China (He
et al., 2020). The dataset integrates a variety of reanalysis, satellite remote sensing and station observation data, and is
widely used in climate change and numerical simulations. The CMFD precipitation data combined with GLDAS non
precipitation field constitute the final driving data of the model through bilinear interpolation.
The initial and boundary conditions for WRF and fully coupled WRF-Hydro model are the Final Operational Global
Analysis (FNL), whose spatial and temporal resolutions are 6 h and 1.00 ° × 1.00 ° respectively
(https://doi.org/10.5065/D6M043C6. Accessed 23 Nov 2022). The vegetation type, land use type, soil type and other land
surface information required by the model are all from the WPS system, and the default soil type is replaced by the soil type
dataset of Beijing Normal University, which has higher accuracy in China. The high-resolution river network data are from
the United States Geological Survey (USGS) Hydrological data and maps based on SHuttle Elevation Derivatives at multiple
Scales (HydroSHEDS) and the resolution of 90.0 m is selected to extract accurate river network information. The details of
the above data are shown in Table 1.
**Table 1.** The overview of the research data.

| Category | Data Type | Spatial/Temporal resolution | Variables |
|---|---|---|---|
| Climate | CMA (V3.0) | 1 d | Precipitation, Temperature, |
| Hydrology | Site | 1 d | Streamflow |
| Eddy Covariance | Site | 30 min | Water/Heat flux, Soil temperature/moisture |
| | CMFD | 3 h; 0.10°×0.10° | Precipitation |
| Driving Data | GLDAS | 3 h; 0.25°×0.25° | Temperature, Wind speed, Solar radiation, Pressure, Specific humidity |
| | FNL | 6 h; 1.00°×1.00° | Initial and boundary conditions |
| Topography | HydroSHEDS | 90.0 m×90.0 m | Digital Elevation Model |





**3 Methodology**
**3.1 Study area**
**3.1.1 WRF model**
The WRF is an advanced and non-hydrostatic mesoscale numerical weather prediction model, which has ability to meet most
mesoscale and small-scale atmospheric and hydrological processes numerical simulation research (Skamarock et al., 2008).
The WRF-ARW model (version 4.1.2) is used for both the WRF and fully coupled WRF-Hydro model in this research.
**3.1.2 Noah-MP model**
The Noah-MP model was developed from the Noah LSM with several major improvements (Niu et al., 2011; Yang et al.,
2011). It provides multiple parameterization options for the key biogeophysical processes, including a separate vegetation
canopy, a two-stream radiation transfer approach and a short-term dynamic vegetation scheme. Besides, the frozen soil
scheme for the groundwater model and the snow model was also updated, which have significant impact on streamflow
simulation (Niu and Yang, 2006). Noah-MP model is selected as the land surface process module of the WRF and the WRF-
Hydro model in this research.
**3.1.3 WRF-Hydro model**
The WRF-Hydro model, developed as a hydrological extension package for WRF, is a new generation of distributed
hydrometeorological forecasting system with physical basis, multi-scale and multi-parameter schemes. It takes the LSM
(Noah/Noah- MP) as a bridge, which connects the large-scale regional climate model with the refined hydrological model.
The model improves the land surface hydrological process related to the spatial redistribution of land surface water,
groundwater and river water, and has capability in quantitatively studying the water-heat exchange process between the
atmosphere and land surface (Gochis et al., 2020). The WRF-Hydro model mainly includes five modules, namely surface
overland flow, saturated subsurface flow, channel, reservoir routing and conceptual baseflow module. The process of
subsurface flow calculates a quasi-3D flow, which takes the vertical and horizontal water exchange into account. The WRF-
Hydro system version 5.1.1 is applied in this research and the complete description of the model is available in Gochis et al.

157 (2020).

**3.2 Experimental designs**
**3.2.1 The parameterization schemes of the WRF and coupled WRF-Hydro model**
The Lambert Projection is adopted in the model with the central longitude and latitude of 99.50°E and 33.75°N and a two-
way nested domains are considered, whose horizontal resolutions of are 25 km and 5 km respectively, as displayed in Fig. 1a.
The vertical structure of both domains consists of 40 levels, up to a 50 hPa pressure top with a time step of 100 s in the outer



domain. The initial and lateral atmospheric boundary conditions for continuous runs are given by the FNL data (in table 1)
which are provided by National Centers for Environment Prediction (NCEP). The physics parameterization schemes
employed in this research for the selected WRF domains are listed in the Table 2, in which the cumulus parameterization is
only used in the outer domain (Senatore et al., 2015). It is important to note that the routing processes with a resolution of
500.0 m are only executed on the innermost domain in the fully coupled WRF-Hydro model. The simulation starts from
March 1st, 2013 to September 1st, 2013 UTC with the first two months as the spin-up time and the rest for analysis.
**Table 2.** Physical options of WRF and the fully coupled WRF-Hydro model.

| Physics process | Parameterization | Reference |
|---|---|---|
| Microphysics | Thompson | Thompson et al. (2008) |
| Cumulus parameterization | Grell-Devenyi (GD) | Grell and Devenyi (2002) |
| Planetary boundary layer | MYNN2 | Nakanishi and Niino (2006) |
| Land surface | Noah-MP | Niu et al. (2011) |
| Longwave radiation | RRTMG | Iacono et al. (2008) |
| Shortwave radiation | RRTMG | Iacono et al. (2008) |

**3.2.2 The parameterization schemes of the WRF and coupled WRF-Hydro model**
Before analyzing the effects of the land-hydrological processes on the atmosphere simulation, WRF-Hydro model is run in
an offline/uncoupled way with the aim of calibrating relevant sensitive parameters and evaluating the performance of the
model in simulating streamflow.
The parameters of the hydrological models are able to reflect the underlying surface characteristics of the region and there
are significant discrepancies in the applicability of the default parameters in models over different basins. In terms of the
WRF-Hydro model, most studies have divided the sensitivity parameters for controlling streamflow process into those for
controlling streamflow distribution and water volume and those for controlling flood peak and flood hydrograph (Gu et al.,
2021). A stepwise manual approach is adopted in calibrating the sensitive parameters, following previous WRF-Hydro
studies (Yucel et al., 2015). It is important to note that the slope of the SRYR is steep, which is different from that of the
Daihe River Basin (Wang et al., 2021). Therefore, the surface retention depth (RETDEPRTFAC) is set as 0.0, and only four
parameters listed in Table 3 are to be calibrated.








**Table 3.** The sensitive parameters of streamflow formation in WRF-Hydro model.

| Classification | Parameter name | Default | Range |
|---|---|---|---|
| Water volume | SMCMAX | - | 0.6~1.2 times |
| | REFKDT | 3.0 | 0.1~5.0 |
| Hydrograph | MannN | - | 0.3~2.0 times |
| | OVROUGHRT | 1.0 | 0.0~1.0 |

Additionally, the water-heat exchange process is of vital importance to the understanding on the atmosphere-land-hydrology process, which has ability to affect the land-surface water cycle process by influencing the evapotranspiration process. Relevant studies show that the default parameterization schemes of Noah-MP model underestimated latent heat (LE) and overestimated sensible heat (H) over the alpine grassland area (Ye et al., 2017). The Chen97 scheme for sensible heat transfer coefficient is able to overcome the problem on overestimating H, while the Jarvis canopy stomatal resistance scheme effectively increases the transpiration of vegetation, so as to improve the simulated LE, and make the distribution of heat flux between LE and H more reasonable. Table 4 shows the parameterization options of Noah-MP used in this research.

**Table 4.** Parameterization scheme options of Noah-MP used in this research.

| Physical Process | Option |
|---|---|
| Dynamic vegetation | Use table LAI; use maximum vegetation fraction |
| Canopy stomatal resistance | Jarvis |
| Soil moisture factor for stomatal resistance | Noah |
| Runoff and groundwater | Original surface and subsurface runoff |
| Surface layer drag coefficient | Chen97 |
| Supercooled liquid water | No iteration |
| Frozen soil permeability | Linear effects, more permeable |
| Radiation transfer | Two-stream applied to vegetated fraction |
| Snow surface albedo | CLASS |
| Rainfall and snowfall | Jordan |
| Lower boundary of soil temperature | Original Noah |
| Snow and soil temperature | Semi-implicit, but FSNO for TS calculation |
| Surface resistent to evaporation/sublimation | Sakaguchi and Zeng for non-snow, rsurf = rsurf_snow for snow |
| Glacier treatment | Slab ice |



**3.3 Evaluation index**
In order to evaluate the model performance of the simulations, Nash Efficiency Coefficient (NSE), Root Mean Square Error
(RMSE), Correlation Coefficient (R) and Relative Deviation (BIAS) are selected in this research. The calculation formula
and optimal value of each evaluation index are shown in Table 5.
**Table 5.** The evaluation indices for simulation performance.

| Indices | Calculation formula | Optimal value |
|---|---|---|
| Correlation Coefficient | $R = \sum_{i=1}^{N} (S_i - \bar{S})(O_i - \bar{O}) / \sqrt{\sum_{i=1}^{N} (S_i - \bar{S})^2 \sum_{i=1}^{N} (O_i - \bar{O})^2}$ | 1 |
| Root Mean Square Error | $RMSE = \sqrt{\sum_{i=1}^{N} (S_i - O_i)^2 / N}$ | 0 |
| Nash Efficiency Coefficient | $NSE = 1 - \sum_{i=1}^{N} (S_i - O_i)^2 / \sum_{i=1}^{N} (O_i - \bar{O})^2$ | 1 |
| Relative Deviation | $BIAS = \sum_{i=1}^{N} (S_i - O_i) / \sum_{i=1}^{N} O_i \times 100\%$ | 0 |


where $N$ is the number of samples, $S_i$ and $O_i$ represent simulated and observed values respectively.
**3.4 Analysis on the applicability of the WRF-Hydro model**
After 2 months spin-up time, the uncoupled WRF-Hydro is calibrated for the period from 1st, June to 1st, September in 2012
based on daily streamflow in Tangnaihai hydrological station. Fig. 2 exhibits that the simulated streamflow is close to the
observation, and the flood hydrograph is consistent with the precipitation hydrograph. For the calibration period, the R of
simulated and measured streamflow is 0.84, the NSE is 0.44, the RMSE is 465.61 $m^3 \cdot s^{-1}$, and the BIAS is -11.44%. For the
validation period, The R is 0.81, the NSE is 0.61, the RMSE is 351.36 $m^3 \cdot s^{-1}$, and the BIAS is -10.21%. However, the
simulated streamflow underestimates the peak flow in flood season and presents some unrealistic peak flows which suggests
the oversimplified base flow model and the uncertainty of the forcing precipitation (Senatore et al., 2015). Nevertheless, the
WRF-Hydro model has ability to produce realistic hydrological regime over the SRYR. Therefore, the calibrated parameters
are used in the comparison between the WRF and the fully coupled WRF-Hydro simulations.

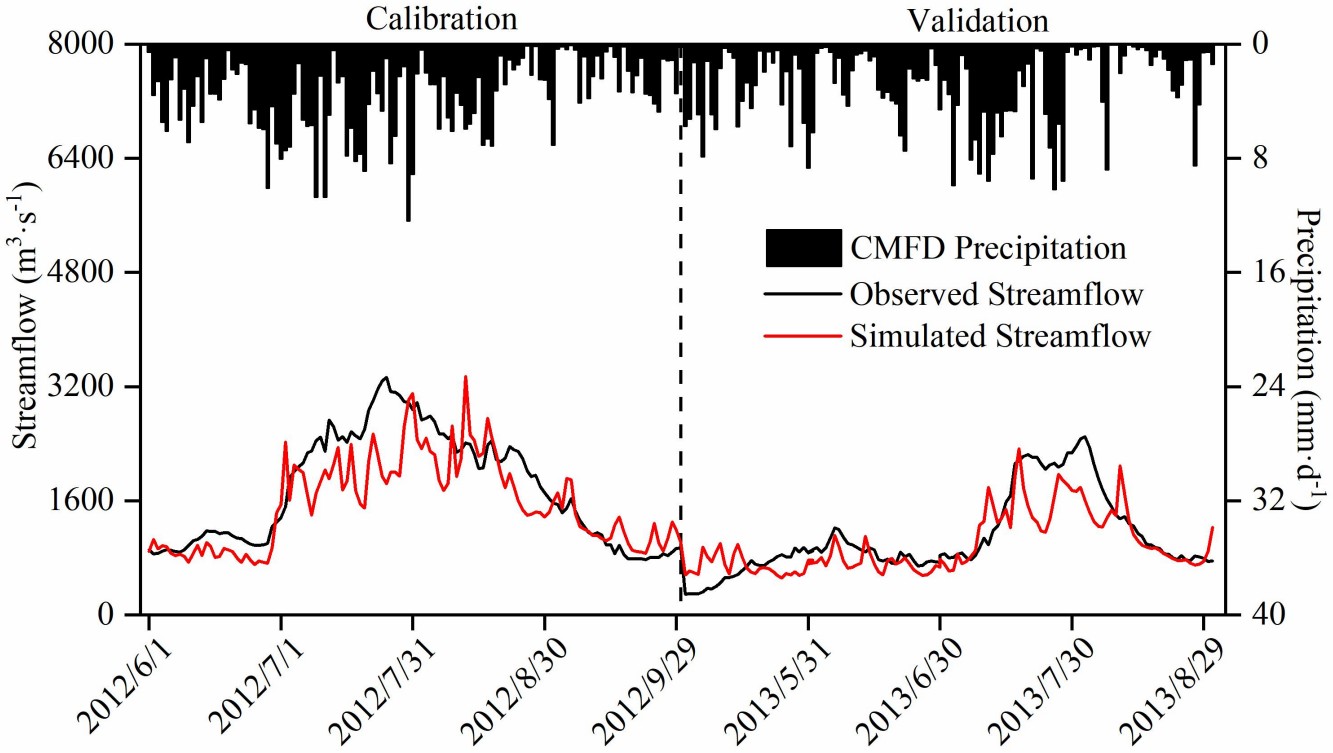

**Figure 2.** The variation of simulated and observed daily streamflow (units: $m^3 \cdot s^{-1}$) over the Source Region of the Yellow River during the calibration and validation period. Black dotted line in correspondence of September 30[th], 2012 splits calibration and validation periods.

Besides, the LE and H fluxes simulated by the uncoupled WRF-Hydro model are also discussed. In order to ensure data comparability, the typical sunny/cloudy days at Eling Lake (lakeside underlying surface) and Maqu (grass underlying surface) stations are selected by characteristic downward total radiation patterns from the entire simulation period. The screening methods of typical sunny/cloudy days are consistent with Zhang et al. (2022).

Fig. 3 and Fig. 4 show the diurnal variation characteristics of LE and H fluxes at two typical flux stations over the SRYR on the typical sunny and cloudy days. On the typical sunny day, the simulated LE is in good agreement with the observation, with the R of 0.92 and 0.96 respectively, and the RMSE of 24.71 $W \cdot m^{-2}$ and 26.89 $W \cdot m^{-2}$. However, there are some differences in the simulation performance of H between the two stations. The WRF-Hydro model has ability to better simulate the diurnal variation of H at the Ealing Lake station with the R of 0.92 and the RMSE of 15.21 $W \cdot m^{-2}$, while underestimates H at Maqu station with the RMSE of 30.29 $W \cdot m^{-2}$.

Due to the complexity of climate conditions on the typical cloudy day, the consistency of simulations and observations is not as good as that on the typical sunny day. On the typical cloudy day, the R between the simulated LE and the observation at two stations is above 0.65 and the RMSE is larger than that of typical sunny days, which is 34.77 $W \cdot m^{-2}$ and 31.30 $W \cdot m^{-2}$, respectively. The simulated H is unable to capture the multi-peak characteristics of H, especially in the Eiling Lake station

with the R of 0.37, while the RMSEs are only 23.66 W·m⁻² and 20.25 W·m⁻². Overall, the WRF-Hydro model has ability to
represent the water-heat exchange process between land and atmosphere over the SRYR.

**Figure 3**. The turbulent fluxes (units: W·m⁻²) simulated by uncoupled WRF-Hydro model on the typical sunny days at Eling

Lake (a-b) and Maqu (c-d) station.Where R represents the correlation coefficient and RMSE is the Root Mean Square Error.

**Figure 4.** As in Figure 3, but for the typical cloudy days.

## 4 Results

Land surface hydrological cycle is one of the important processes in the Earth System. On the one hand, climate change drives the water cycle process at the global scale and causes different responses at the regional scale. On the other hand, the variations of land surface process further alter the distribution of water resources and runoff at regional and catchment scales (Meng et al., 2020). Based on the fully coupled WRF-Hydro model, the impact of climate change on land surface hydrological processes and the feedback of surface water cycle to precipitation are comprehensively considered, and the variation characteristics of the coupled process of atmosphere-land-hydrology and streamflow over the SRYR in the rainy season (May-August) of 2013 are also explored.



## 4.1 The validation of temporal variation of hydrometeorological elements

The influence of global climate change on hydrological environment is extremely significant, in which temperature and precipitation are the key factors to the redistribution of water cycle in time and space (Meng et al., 2020). Precipitation directly participates in all aspects of the water cycle, while temperature also indirectly influences the whole process of the water cycle through evaporation and snowfall. Therefore, the simulation performance is first evaluated by comparing precipitation and temperature from WRF and coupled WRF-Hydro simulations with observations.

Fig. 5 exhibits the Taylor diagrams (Taylor, 2001) for daily precipitation and temperature among simulations with observations of 9 meteorological stations. The simulated precipitation is in good agreement with the observations, with the correlation coefficients higher than 0.6. The standardized deviation ratios of the coupled WRF-Hydro to the observations are between 0.8 and 1.2 with greater RMSE than the standalone WRF model, which means that the coupled process increases the wet bias of precipitation simulation since considering the terrestrial lateral flow of soil water. In terms of temperature, both two experiments perform well in simulating temperature, with the average R is 0.93 and a slight cold deviation.

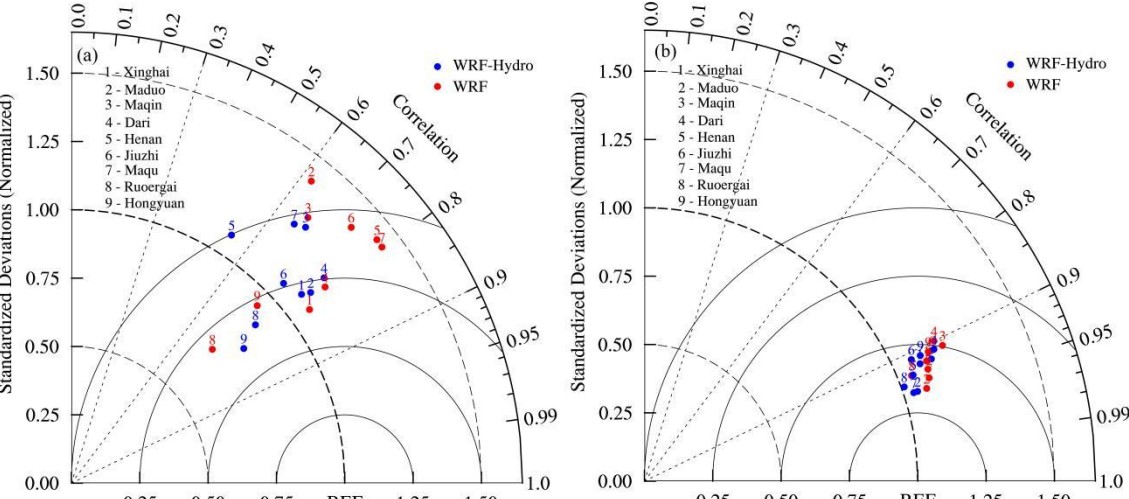

**Figure 5.** Taylor diagrams of correlation coefficients and standard deviations for daily precipitation and temperature over the SRYR among simulations with observations from May 1st, 2013 to August 31st, 2013.

Since the sparse and uneven distribution of stations over the SRYR, the reanalysis products are employed to analyze the spatial variation of the hydrometeorological elements in this research. Relevant researches (Li et al., 2014; Chen et al., 2022) indicate that CMFD precipitation data have good applicability over the SRYR, and climate and hydrological factors such as temperature and runoff of GLDAS have ability to better characterize the climate change and water cycle processes over the SRYR. Therefore, CMFD and GLDAS are taken as reference (denoted as Reference) datasets to compare and analyze the regional mean simulation results in the following study.





Fig. 6 shows that the time series of regional mean meteorological and hydrological elements. It exhibits that both WRF and
fully coupled WRF-Hydro model are able to characterize the evolution characteristics of meteorological and hydrological
elements over the SRYR. Compared with standalone WRF model, the coupling process slightly increases the wet deviation
of precipitation, while it improves the simulation of temperature and downward longwave and shortwave radiation to a
certain extent. The simulations of surface pressure, specific humidity of 2m and wind speed near the ground have little
difference in two experiments. However, the coupling process increases soil moisture due to considering of the terrestrial
vertical and lateral flow of soil water in three-dimensional space, so the simulation results of evapotranspiration are larger.





**Figure 6**. The time series of mean meteorological and hydrological elements simulated by WRF and fully coupled WRF-
Hydro model from the period of May 1$^{st}$, 2013 to August 31$^{st}$, 2013.
The land-atmosphere water and heat exchange processes affect the surface soil moisture by changing evapotranspiration and
further impacts the surface energy and hydrological cycle process (Jia et al., 2014). The diurnal variation characteristics of
the LE and H simulated by WRF and coupled WRF-Hydro model on the typical sunny days, when the land-atmosphere
water and heat exchange process is relatively strong, are analyzed in Fig. 7. The results at Eling Lake station show that the
LE and H simulated by fully coupled WRF-Hydro model are more consistent with the measured, with the RMSE reducing to
29.91 W·m$^{-2}$ and 31.06 W·m$^{-2}$ compared with WRF simulations. For the Maqu station, the coupled process increases the
deviation of LE simulations, which is related to the overestimation of evapotranspiration, while the problem that H are
overestimated is greatly improved.

**Figure 7.** Comparison of turbulent fluxes (units: W·m$^{-2}$) between WRF-Hydro/ WRF and observations on the typical sunny
days at Eling Lake (a-b) and Maqu (c-d) station.





Fig. 8 demonstrates the diurnal variation characteristics of the LE and H on the typical cloudy days, when the weather
conditions and physics processes are more complicated. On the typical cloudy days, the agreement of the turbulent fluxes
simulation results with observations is not as good as that on the typical sunny day. The simulations at two stations indicate
that coupled process is able to reduce the RMSE in LE and H with a mean RMSE of 29.48 $W \cdot m^{-2}$ and 26.55 $W \cdot m^{-2}$
respectively.
Overall, the coupled WRF-Hydro simulations improve the simulation of surface heat flux variables duo to the consideration
of the lateral terrestrial water flow of hydrological process.


**Figure 8**. As in Figure 5, but for the typical cloudy days.

Soil temperature and moisture are able to affect the land-surface evaporation and groundwater processes, directly or
indirectly affect the land-hydrology process. Therefore, the top-layer soil temperature and moisture at Maqu station are
analyzed in Fig. 9 due to the availability of observed data. The results display that the soil temperature simulated by WRF





and coupled WRF-Hydro model is in good agreement with observation. However, the simulated soil temperature is far
greater than the observed in August, which is related to the deviations from the downward shortwave radiation and
temperature (Fig. 6). The coupling process reduces the RMSE of soil temperature simulation (from 5.18 to 4.22 K) for the
comprehensive consideration in the variation of soil water content. Besides, WRF-Hydro maintains a longer soil moisture
memory with respect to the standalone WRF run and the simulated soil moisture values from WRF-Hydro significantly
exceed WRF ones (Fig. 9b), which is due to the subsurface lateral flow considered in WRF-Hydro model. The simulated top-
layer soil moisture of both experiments is not consistent with the observation and has difficulty in reflecting the response of
soil moisture to precipitation. Additionally, the comparison of site results also brings uncertainty to the judgment of
simulation performance which needs to be verified in spatial distribution.





**Figure 9.** The time series of top-layer soil temperature (units: K) and moisture (units: m³/m³) simulated by WRF and fully
coupled WRF-Hydro model from the period of May 1st, 2013 to August 31st, 2013.
**4.2 The validation of spatial distribution of hydromteorological elements**
Furthermore, the spatial distribution of the accumulated precipitation for CMFD reference data (denoted as Reference), WRF
and coupled WRF-Hydro, as well as their differences are displayed in Fig. 10. The precipitation highlights the strong
dependence of rainfall patterns on topography and shows a decreasing trend from southeast to northwest over the SRYR.
The WRF and fully coupled WRF-Hydro have ability to better capture the distribution characteristics of precipitation. The
simulations of both experiments have a significantly wet bias, especially in the Jiuzhi and Maqu area, while a dry bias in the
southeastern SRYR. Compared with the standalone WRF, the coupled WRF-Hydro takes the subsurface lateral flow into
consideration, which leads to the increase of soil moisture and a more reasonable spatial distribution of soil water, so as to
have a feedback effect on precipitation with a mean wet bias of 16.63 mm than WRF simulations.





**Figure 10.** The spatial distribution of the accumulated precipitation (units: mm) in the time interval May 1st, 2013 to August 31st, 2013 with (a) observation, (b) the standalone WRF simulation, (c) the coupled WRF-Hydro simulation, and difference map for (d) WRF minus observation, (e) WRF-Hydro minus observation, (e) WRF-Hydro minus WRF.

With the respect to temperature, the spatial patterns of the mean 2-m air temperature during the simulated period are showed in Fig. 9. The spatial distribution of temperature presents gradient characteristics with higher temperatures in the flat regions and lower temperatures in the alpine areas. Both experiments have ability to capture the distribution characteristics of temperature well. On the whole, the simulated temperature is relatively higher, especially in the northeast of the SRYR, with a regional mean bias of 0.44 K for standalone WRF and of 0.15 K for the fully coupled WRF-Hydro respectively, which means that the coupled simulation slightly reduces the deviation of the mean temperature.

**Figure 11.** As in Figure 8, but for the mean temperature (units: K).





The spatial distribution of turbulent fluxes is able to affect the development of the atmospheric boundary layer through
thermal exchanges, and then change the precipitation structure and area (Zhang et al., 2019). Fig. 12 shows the spatial
distribution of the LE and H simulated by WRF and coupled WRF-Hydro models and their differences. As far as LE is
concerned, both experiments demonstrate a southeast-northwest gradient with two maximum value areas of LE in Zhaling
Lake and Eiling Lake, which is related to the evaporation process of lakes. Compared to standalone WRF (mean value 55.89
$W \cdot m^{-2}$), the coupled simulation (mean value 68.98 $W \cdot m^{-2}$) increases the LE simulation values due to the increase in
evaporable area. In terms of H, its spatial distribution is the opposite of LE, showing decreasing characteristics from
northwestern to southeastern SRYR, with a low value area for simulated H in the two lakes. The coupled simulation
significantly reduces the simulated H and overcomes the problem of large H simulation, with a mean difference of -4.2
$W \cdot m^{-2}$ between the coupled WRF-Hydro and WRF.

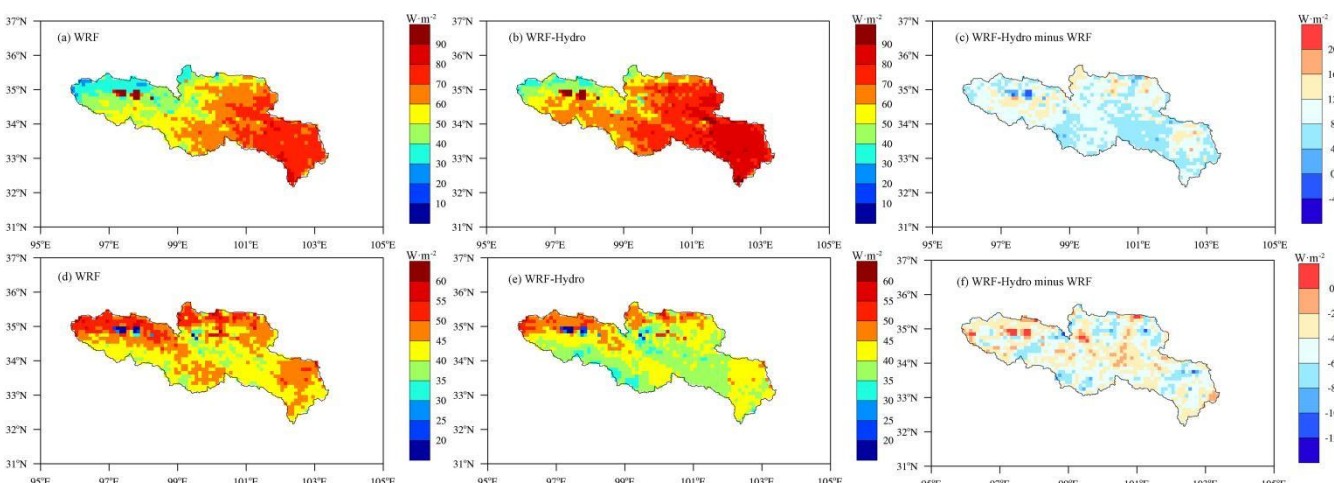


**Figure 12.** The spatial distribution of the mean latent heat (units: $W \cdot m^{-2}$) in the time interval May 1st, 2013 to August 31st,
2013 with (a) the standalone WRF simulation, (b) the coupled WRF-Hydro simulation, and difference map for (c) WRF-
Hydro minus WRF, (d-f) are same as (a)-(c), but for sensible heat (units: $W \cdot m^{-2}$).
Soil temperature is an important parameter of land surface process, which directly reflects the land thermal state. Its variation
affects the movement and phase transition of surface soil water, thus affecting the surface hydrological cycle (Zhang et al.,
2021). Soil moisture is a key factor affecting land-atmosphere interaction. As a storage term for heat and moisture, soil
moisture has the ability to remembering from weeks to months, remembering previous atmospheric perturbations and then
influencing the atmosphere through factors such as surface fluxes (Song et al., 2019).
The spatial distribution characteristics of top-layer soil temperature (Fig. 13a, b, c) and moisture (Fig. 13d, e, f) during the
2013 rainy season over the SRYR are analyzed in Fig. 13. Due to the high altitude and the large temperature difference





between day and night over the TP, its mean temperature is lower than that of the inland areas, so the closer to the depth of
the plateau, the lower the soil temperature. Both experiments are able to better depict the characteristic that the temperature
in the lake area is lower than the surrounding area, and the coupled WRF-Hydro reduces the simulated surface soil
temperature, with a cold deviation of 1.07 K, which is able to affect the atmospheric water vapor convergence through land-
atmosphere interactions. For soil moisture, both experiments show that wet centers in the Zaling Lake and Eling Lake area.
Influenced by the terrestrial lateral water and soil moisture redistribution process of coupled WRF-Hydro, the spatial
distribution of soil water content is more reasonable over the study region, and WRF-Hydro soil moisture values
significantly exceed WRF ones, with a wet deviation of 0.02 $m^3/m^3$. On the whole, the two lakes and the surrounding areas
over the SRYR are cold and wet centers during the simulation period, and the coupled simulations better capture the
variation characteristics of soil temperature and moisture.

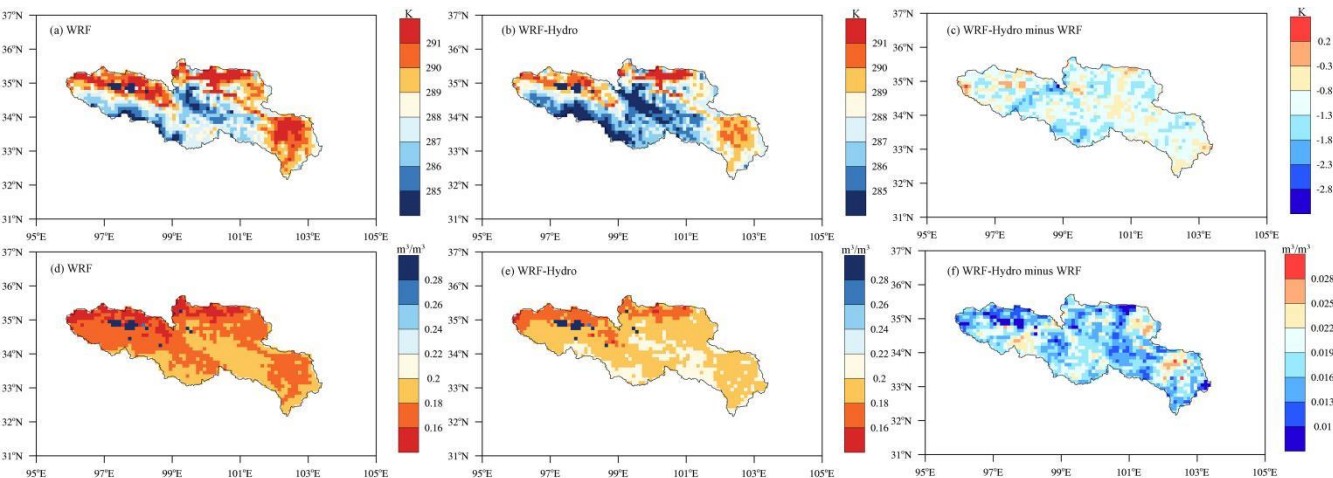

**Figure 13.** The spatial distribution of the mean top-layer (0-10 cm) soil temperature (units: K) in the time interval May 1st,
2013 to August 31st, 2013 with (a) the standalone WRF simulation, (b) the coupled WRF-Hydro simulation, and difference
map for (c) WRF-Hydro minus WRF, (d-f) are same as (a)-(c), but for soil surface temperature (units: $m^3/m^3$).
**4.3 The time series of the streamflow simulated by the fully coupled model**
The time series of fully coupled simulated streamflow for the 2013 rainy season when no direct meteorological station
observations are required are given in Fig. 14. The fully coupled model is able to capture the temporal variation of observed
hydrographs with a R of 0.77. However, reproducing the daily streamflow with the fully coupled model still remains a
challenge with a NES of 0.33 and a RMSE of 458.85 $m^3 \cdot s^{-1}$. The main reason is that the coupled simulated streamflow is
severely overestimated, especially the reproduction of peak flow is very limited. The performance degradation is mainly due
to the fact that the WRF-Hydro model is extremely sensitive to the quality of precipitation data, where the RMSE of
precipitation is only 2.51 mm, but the RMSE of streamflow reaches 458.85 $m^3 \cdot s^{-1}$. Besides, this error may also be due to the



different frequencies of the Noah-MP LSM called in the uncoupled calibration and fully coupled runs. During the uncoupled
simulations, the Noah-MP LSM is typically invoked at the physical time step of the hydrological model, whereas during
fully coupled simulations, it is invoked at the physical time step of the WRF model, which results in more water passing
farther downslope or into the channel before infiltration occurs again, implying higher streamflow values (Senatore et al.,
383 2015).

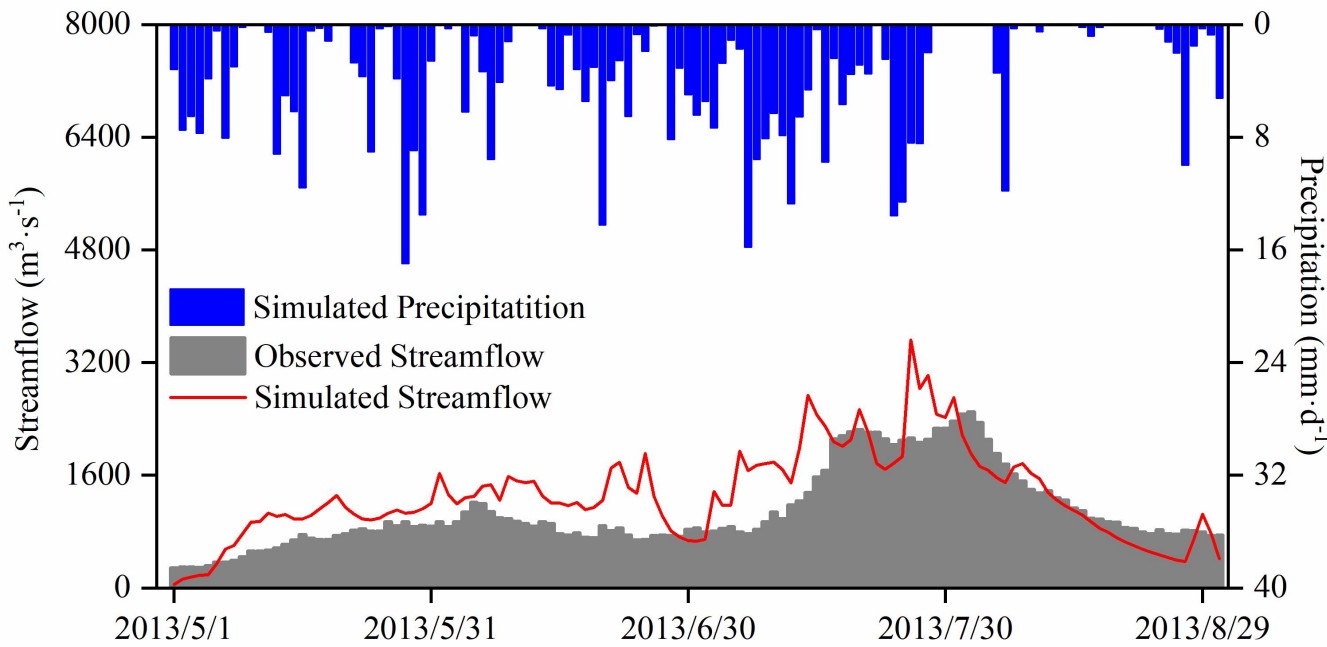


**Figure 14.** The observed and fully coupled WRF-Hydro simulated streamflow (units: $m^3 \cdot s^{-1}$) for the period May 1st, 2013 to
August 31st, 2013.

## 5 Discussion

The SRYR is rich in wetland resources and is the most important water conservation area in the middle and upper reaches of
the Yellow River, and the contribution of groundwater to surface runoff cannot be ignored (Jia et al., 2022). In this research,
the streamflow simulated by the uncoupled WRF-Hydro model is generally in good agreement with the observation.
However, the detailed portrayal of the hydrograph needs to be improved, especially the steep change of streamflow at the
flood peak, which does not well reflect the modulating effect of lateral flow of groundwater on streamflow changes.
Therefore, in order to clarify the mechanisms of groundwater and soil water content on streamflow, a comprehensive
analysis of land surface water cycle processes is supposed to be conducted in further research.
In addition, there is a significant overestimation of streamflow simulated by the fully coupled model in this research. To
better investigate the possible reasons for the degradation of the coupled simulation performance, the sensitivity analysis





plots of different atmospheric driving data combinations on the simulated streamflow from the uncoupled WRF-Hydro
model are given in Fig. 15. The quality of precipitation data is very sensitive to streamflow simulations (WRFOUT) relative
to the initial combination of driving data (GLDAS+CMFD) in this research, while non-precipitation data do not have a
significant impact on streamflow (WRFOUT+CMFD). When using fully coupled simulated precipitation data, the slight
deviations in precipitation (RMSE of 2.51 mm) cumulatively lead to larger error (458.85 $m^3 \cdot s^{-1}$) in the streamflow simulation.
Therefore, reproducing daily streamflow with the fully coupled model remains a challenge at present (Li et al., 2021).
Studies have shown that data assimilation has ability to improve the effect of precipitation forecasting at small and medium
scales, promote the development of coupled atmosphere-land-hydrology simulation level, and is expected to further improve
the accuracy of flood forecasting on the basis of extending the forecasting period (Gu et al., 2021). Therefore, satellite and
radar data are able to be assimilated into the WRF model in the future research to improve the simulation results of the
precipitation and streamflow. Meanwhile, the work of Senatore et al. (2015) is contribute to addressing the issue of the effect
of different frequencies that the LSMs being called on streamflow simulation by enabling multiple time steps for different
forcing and model components in the WRF-Hydro model.

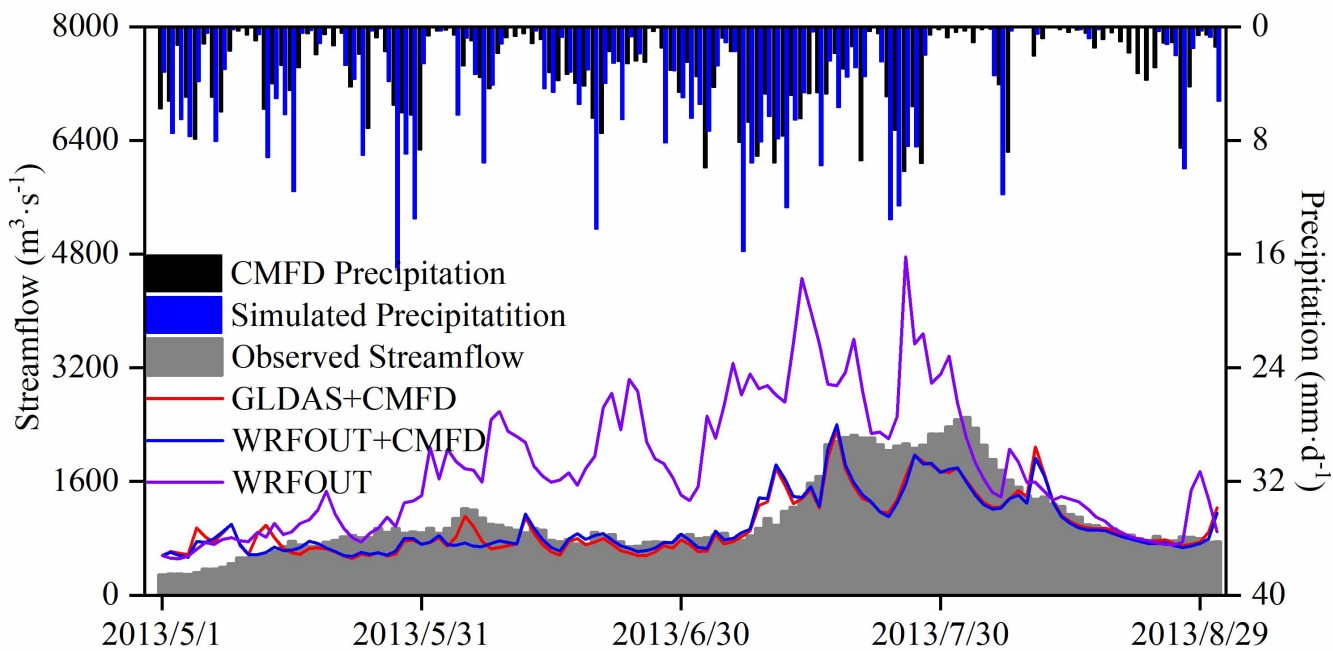


**Figure 15.** The uncoupled WRF-Hydro simulated streamflow (units: $m^3 \cdot s^{-1}$) driven by different precipitation data for the

period May 1$^{st}$, 2013 to August 31$^{st}$, 2013.



## 6 Conclusions


Based on WRF-Hydro model, the meteorological, hydrological, eddy covariance stations and multiple reanalysis data over
the SRYR, the key variables of the coupled atmosphere-land-hydrological processes simulated by standalone WRF and
coupled WRF-Hydro model during the 2013 rainy season (May-August) are compared and analyzed to investigate the effects
of climate change on land surface and water cycle processes and the feedback of land surface hydrological cycle to
precipitation over the SRYR in this research. The following conclusions have been drawn:
1) The uncoupled WRF-Hydro model is able to characterize the variability of streamflow over the SRYR basin with a R of
0.84 and a NSE of 0.44 for the simulated streamflow and observations during the calibration period, and a R of 0.81 and a
NSE of 0.61 for the validation period.
2) In terms of temporal variation, both the results of the standalone WRF and the coupled WRF-Hydro model indicate the
reasonable performance in reproducing variables of atmospheric-land-hydrological processes over the SRYR. Compared
with WRF model, the coupled process improves the simulation results for temperature, downward longwave and shortwave
radiation, but slightly increases the wet bias of the precipitation and evapotranspiration, with a RMSE increased from 2.50 to
2.51 mm for precipitation and from 0.71 to 0.78 mm for evapotranspiration. Due to the consideration of lateral flow of soil
water, the coupling process significantly reduces the bias of the simulation results of water-heat exchange fluxes, soil
temperature and moisture, with a mean RMSE of 32.27 W·m$^{-2}$, 24.91 W·m$^{-2}$, 4.22 K, and 0.06 m$^3$/m$^3$ respectively.
3) In terms of spatial distribution, the coupled simulation increases the wet bias of the precipitation with a mean wet bias of
16.63 mm than WRF simulations, which is caused by the lateral redistribution and reinfiltration of soil water, but slightly
enhance the simulation results of temperature. The coupled process increases the LE and overcomes the problem of large H
simulation, and also results in a slight wetting and cooling of the near-surface atmosphere, making the spatial distribution of
water-heat exchange flux and soil temperature and moisture more reasonable.
4) The fully coupled model is able to capture the variation characteristics of streamflow. However, reproducing daily
streamflow with the fully coupled model remains a challenge, with a NSE of 0.33 and RMSE of 458.85 m$^3$·s$^{-1}$, because of
the uncertainty of the simulated precipitation.

**Code and data availability.** The source code of the WRF-4.1.2 and WRF-Hydro-5.1.1 models are available from
https://www2.mmm.ucar.edu/wrf/users/download/get_source.html (Skamarock et al., 2008) and
https://ral.ucar.edu/projects/wrf_hydro/model-code (Gochis et al., 2020) respectively. The daily streamflow data of
Tangnaihai hydrological station can be downloaded at http://www.yrcc.gov.cn, the observations of the meteorological
stations collected from the National Meteorological Information Center at
http://101.200.76.197/data/cdcdetail/dataCode/SURF_CLI_CHN_MUL_DAY_V3.0.html and he turbulent heat fluxes and



soil temperature and moisture data of the Eling Lake station from the National Cryosphere Desert Data Center

(http://www.ncdc.ac.cn, Meng and Lyu, 2022). The CMFD data are provided by the National Tibetan Plateau Data Center

(https://doi.org/10.11888/AtmosphericPhysics.tpe.249369.fle, He et al., 2020). The GLDAS data are from

https://ldas.gsfc.nasa.gov/gldas/ (Beaudoing and Rodell, 2020).

**Author contributions**. YC and WJ mainly wrote the manuscript and were responsible for the research design, data

preparation and analysis. XM provided critical insights to the research results. And the other authors discussed and assisted

with interpretation of the results and contributed to the article. All authors have read and agreed to the published version of

the manuscript.

**Competing interests**. The authors declare that they have no conflict of interest.

**Acknowledgments.** The authors would like to thank the National Cryosphere Desert Data Center and National

Meteorological Information Center for providing the precious observed data. Moreover, we acknowledge the developers of

the open-source models used for this research.

**Financial support**. This research has been jointly supported by the the Science and Technology Plan Project of Sichuan

Province (Grant 2021YJ0025) and Scientific Research Project of Chengdu University of Information Technology (Grant

KYTZ201821)

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
