# Peer review of "A Study on the Fully Coupled Atmosphere-Land-Hydrology Process and 2 Streamflow Simulations over the Source Region of the Yellow River"

_Hydrology and Earth System Sciences, 2022_

## Author Comment (AC1)

Dear Anonymous Reviewer,

Thanks for your comments for the manuscript entitled "A study on the fully coupled atmosphere-land-hydrology process and streamflow simulations over the Source Region of the Yellow River". These suggestions are quite valuable and helpful to improve the quality of our manuscript. We have carefully read the comments and made necessary corrections or revisions. A point-to-point response to your comments is presented following.

Sincerely.

Jun Wen on behalf all authors

**REVIEWER COMMENTS**

**A very interesting study, I am impressed by the involved simulation experiments. My recommendation is that the paper be published in HESS provided the following points are addressed by the authors.**

Response: Thanks for your recognition and constructive comments to this research, the manuscript was carefully revised with reference to your suggestions.

**Q1. How to prove the conclusion is robust since that the study only focused on the 2013 rainy season. If we look at different periods, whether the results and related conclusion change.**

Response: Thanks for your question, the above-mentioned two questions are to be addressed as following separately.

1) Firstly, the capabilities of WRF-Hydro model in long-term hydrological process simulation over the Source Region of the Yellow River has been proved in author's another research (Chen et al., 2022). Besides, the aim of this study is to investigate the effects of climate change on land surface and water cycle processes and the feedback of land surface hydrological cycle to precipitation. On the basis of the reliable driving data from coupling model, the simulation of the hydrological process is also reliable. Therefore, the study only focuses on the 2013 rainy season is reasonable and the conclusion is robust.

2) Since the calibrated model is capable of characterizing the fully coupled atmosphere-land-hydrology process over the Source Region of the Yellow River, the relevant results may have differences in values when paying attention to different time periods, but within the acceptable range, which will not affect the qualitative conclusions.

**Q2. For the NSE values in calibration and validation periods, it is a little strange that the model has better skill in validation period, which differs from our normally expection. Why?**

Response: Thanks for the questions you raised. In fact, we also noticed this fact during the analyzing the results. By checking the driven data and the sensitivity parameters of the model, as well as consulting the relevant references of daily streamflow simulation over the Source Region of the Yellow River, it can be found that the model is very sensitive to precipitation, the large value of the precipitation data lead to the large simulated streamflow during the calibration period and the appearance of the simulated streamflow peak later with a smaller NSE value, which also appears in other relevant references (Zhang et al., 2017; Gu et al., 2021). However, the variation of precipitation hydrograph during the validation period is relatively gentle and consistent with the streamflow hydrograph. So it could be concluded that the model has somewhat better skill during the validation period.

**Q3. Just as pointed out in the introduction that several researchers had already applied the coupled WRF-Hydro in simulating hydrological process in source regions of the Three River, so, what's the novelty in this study? Are there any research gaps that were still not addressed, or did the author get some new findings?**

Response: Thanks for your comments. Just as pointed out in the introduction session, the researches on the coupled/uncoupled WRF-Hydro in simulating hydrological process mainly focus on the short-term flood events in small and medium-scale watersheds which mostly located in plain areas with the single underlying surface conditions, the calibration of the sensitive parameters is relatively easy. So one of the novelties in this study is that the study region is a large-scale watershed with complex

underlying surface and climate conditions, and there are large challenges in the calibration of the sensitive parameters and the calculation of the model. The other novelty is that the fully coupled atmosphere-land-hydrology process is explored and the water-heat exchange process between the atmosphere and land surface in this research is quantitatively studied which is rarely considered in other studies.

Besides, compared with other studies which focused on the hydrological process over the Source Regions of the Three River by using WRF-Hydro model, this research gets a higher NSE in streamflow simulation and proves that the fully coupled WRF-Hydro model has ability to reproduce the daily streamflow over the Source Region of the Yellow River, which is a large improvement compared with the research results of Li et al (2021).

**Q4. Figure 1, it is suggested to involve much more hydrological observation in this study. Such as the streamflow JiMai, MaQu…**

Response: Thanks for your suggestions. Due to data security and other reasons, it is very difficult to obtain the observed streamflow data. At present, the daily streamflow data over the Source Region of the Yellow River which can be obtained from the Yellow River Water Conservancy Bureau is only Tangnaihai hydrological station. Therefore, the applicability of the model in other hydrological stations might be potentially verified in the future research.

**Q5. Figure 5, please supply the captions for temperature and precipitation in the figure title.**

Response: Thanks for your reminding. The captions for temperature and precipitation have been supplied in the figure title in the revised manuscript. (Line 261, page 13)

**References:**

Chen, Y. L., Wen, J., Yang, C. G., Long, T. P., Li, G. W., Jia, H. J., and Liu, Z.: Analysis on the applicability of different precipitation products and WRF-Hydro model over the Source Region of the Yellow River, Chinese Journal of Atmospheric Sciences, [preprint], https://doi.org/10.3878/j.issn.1006-9895.2205.22057, 2020.

Gu, T. W., Chen, Y. D., Gao, Y. F., Qin, L. Y., Wu, Y. Q., and Wu, Y. Z.: Improved streamflow forecast in a small-medium sized river basin with coupled WRF and WRF-Hydro: Effects of radar data assimilation, Remote Sensing, 13, 3251, https://doi.org/10.3390/rs13163251, 2021.

Li, G. W., Meng, X. H., Blyth, E., Chen, H., Shu, L. L., Li, Z. G, Zhao, L., and Ma, Y. M.: Impact of fully coupled hydrology-atmosphere processes on atmosphere conditions: Investigating the performance of the WRF-Hydro model in the Three River Source Region on the Tibetan Plateau, China, Water, 13, 3409, https://doi.org/10.3390/w13233409, 2021.

Zhang, A., Li, T. J., Fu, W., and Wang, Y. T.: Model simulation of flood season runoff in the headwaters of the Yellow River Basin using satellite-ground merged precipitation data, Journal of Basic Science and Engineering, 25, 1-16, https://doi.org/10.16058/j.issn.1005-0930.2017.01.001, 2017.